# Modeling the Wintering Habitat Distribution of the Black Stork in Shaanxi, China: A Hierarchical Integration of Climate and Land Use/Land Cover Data

**DOI:** 10.3390/ani13172726

**Published:** 2023-08-27

**Authors:** Yilamujiang Tuohetahong, Ruyue Lu, Feng Gan, Min Li, Xinping Ye, Xiaoping Yu

**Affiliations:** 1College of Life Sciences, Shaanxi Normal University, Xi’an 710119, China; ilhambhdr@sina.com (Y.T.);; 2School of Environment and Resources, Taiyuan University of Science and Technology, Taiyuan 030024, China; 3Research Center for UAV Remote Sensing, Shaanxi Normal University, Xi’an 710119, China; 4Changqing Teaching & Research Base of Ecology, Shaanxi Normal University, Xi’an 710119, China

**Keywords:** black stork (*Ciconia nigra*), species distribution models (SDMs), land use/land cover (LULC), climate change, wintering habitats

## Abstract

**Simple Summary:**

Climate and land use/land cover changes are key factors that significantly impact the distribution of wild species. The black stork, an endangered bird with a high conservation status in China, is particularly vulnerable to these changes due to its long-distance migration. Previous studies have mainly focused on the effect of climate change on the black stork’s breeding season. However, little attention has been given to how its wintering habitats are affected by these factors at different scales. In our study, we evaluated the dynamics of the black stork’s wintering habitat distribution in response to global and regional changes. Our findings indicate that both climate and land use/land cover changes significantly impact the distribution of the wintering black stork. However, the impact of human activities is more pronounced. In order to mitigate the negative effects of human-induced land use/land cover changes on the black stork and other migratory birds, it is crucial to reduce the destruction of wildlife habitats caused by human activities. Additionally, we need to focus on effectively protecting the migratory pathways and wintering sites of these birds, including those that are outside the existing protected areas.

**Abstract:**

Species distribution models (SDMs) are effective tools for wildlife conservation and management, as they employ the quantification of habitat suitability and environmental niches to evaluate the patterns of species distribution. The utilization of SDMs at various scales in a hierarchical approach can provide additional and complementary information, significantly improving decision-making in local wildlife conservation initiatives. In this study, we considered the appropriate spatial scale and data resolution to execute species distribution modeling, as these factors greatly influence the modeling procedures. We developed SDMs for wintering black storks at both the regional and local scales. At the regional scale, we used climatic and climate-driven land use/land cover (LULC) variables, along with wintering occurrence points, to develop models for mainland China. At the local scale, we used local environmental variables and locally gathered wintering site data to develop models for Shaanxi province. The predictions from both the regional and local models were then combined at the provincial level by overlapping suitable areas based on climatic and local conditions. We compared and evaluated the resulting predictions using seven statistical metrics. The national models provide information on the appropriate climatic conditions for the black stork during the wintering period throughout China, while the provincial SDMs capture the important local ecological factors that influence the suitability of habitats at a finer scale. As anticipated, the national SDMs predict a larger extent of suitable areas compared to the provincial SDMs. The hierarchical prediction approach is considered trustworthy and, on average, yields better outcomes than non-hierarchical methods. Our findings indicate that human-driven LULC changes have a significant and immediate impact on the wintering habitat of the black stork. However, the effects of climate change seem to be reducing the severity of this impact. The majority of suitable wintering habitats lie outside the boundaries of protected areas, highlighting the need for future conservation and management efforts to prioritize addressing these conservation gaps and focusing on the protection of climate refuges.

## 1. Introduction

Global and regional ecosystems are facing serious threats due to global change [1,2,3,4]. Multiple studies have demonstrated that global change has had a considerable impact on species, populations, and ecosystems, and this impact has only intensified over time [5,6]. Climate change is a well-known cause of habitat loss, which alters species’ distribution patterns and their occupied ranges. In addition to climate change, changes in land use and land cover (LULC) and the resulting habitat fragmentation can significantly affect biological processes and reduce the quantity and quality of habitats [2]. Studies have employed climate-driven simulations at various scales, research regions, and species to estimate the possible effects of climate change on species distribution [1,7,8,9]. Furthermore, other studies have explicitly addressed climate-LULC interactions [10,11,12]. However, very few studies paid attention to the spatial scale of climate and LULC variables [13,14]. The appropriate spatial scale and data resolution (or “cell size”) are critical factors in the execution of species distribution modeling. It has been suggested that species distribution is mostly affected by climate at the macroscales, thus requiring broader spatial extents and coarser data resolutions to effectively establish correlations between climate and species distributions [15]. It has also been hypothesized that LULC may be the predominant factor in determining the presence and absence of species at a finer spatial resolution compared to climate [15,16]. However, LULC changes can be influenced by various factors, including human activities, terrain features, and climate. While human activities affect LULC on a smaller scale, climate change impacts LULC on a larger scale with a coarser resolution. To better understand the habitat suitability of species in changing environments at different scales, it is crucial to consider both the LULC associated with human activities and the LULC changes caused by climate change. Therefore, previous studies [17,18] have proposed species distribution models (SDMs) calibrated at different scales to account for varied scale-dependent processes driving habitat distributions. The black stork (*Ciconia nigra*, Linnaeus, 1758), which can utilize various types of wetlands and undergo long-distance migrations, can provide an appropriate study example for the research mentioned above. 

The black stork is a large wading bird that is widely distributed in the Palearctic region [19]. This species is notable for its long-distance migration between Europe and Africa or West Asia and India [20]. In China, the black stork can be found in almost every province except for Tibet, both in its breeding and nonbreeding periods. During winters, the species primarily inhabits several provinces, including Shanxi, Henan, Shaanxi, Sichuan, Yunnan, Taiwan provinces, and the Yangtze River basin. Wetlands in the midstream of the Yellow River, the Weihe River, and its tributaries, and the Hanjiang River basin are the primary wintering habitats in Shaanxi Province. Similar to other migratory birds, the black stork is vulnerable to the impact of climate change on regions along their migratory routes [21,22]. However, assessing the potential effect of climate change on this species is challenging due to its complex migration patterns [23]. Although many trans-Saharan species are predicted to reduce and shift their wintering range as a result of climate change [24], little is currently known about the black stork’s wintering habitats in the Palearctic region [22]. 

In this study, we propose two ecological and spatial scales, national and provincial, to effectively capture the fundamental factors contributing to habitat distribution patterns [10,17] and generate two different but complementary models, aligning with current advancements in ecological modeling and climate change projections. There are two main reasons why SDMs calibrated at both the national and provincial scales are considered important. Firstly, national-scale SDMs are expected to take into account the broader ecological factors and their variations, such as climate change and climate-driven LULC changes, as these factors operate on a larger scale and can greatly influence species distribution patterns. Secondly, provincial-scale SDMs are essential in capturing the more localized variables with finer resolution that affect species distribution, including topography, human disturbance, water and food availability, and LULC. However, it should be noted that SDMs calibrated across small geographic areas, such as at the provincial scale, may not include the complete range of climatic niches required by certain species, particularly migratory birds. Climate change projections generated from these models may produce biased future species distribution predictions compared to models calibrated at larger scales [13]. Therefore, it is not appropriate to project models calibrated at smaller scales to future climate scenarios. Instead, using a larger scale, such as a continental or national level, can cover a higher proportion of the climatic condition of the species and provide accurate predictions for future climate change. Consequently, this study will only utilize SDMs calibrated at the national scale for projecting into future climate scenarios.

Based on previous research in other areas [16,25], we hypothesize that the species distribution models obtained at the national and provincial scales will be different. These models represent different scale-dependent drivers of species distribution patterns and can provide complementary information for the wintering habitat of the black stork. National models are expected to provide details on the distribution of wintering habitats based on a larger proportion of climate niches and accurate future climate projections, which will help in generating precise spatial forecasts of the black stork’s future climate niche. In contrast, provincial models are anticipated to capture how wintering habitats are distributed under certain climate conditions and, thus, represent important local conditions for wintering habitat [17]. Therefore, a hierarchical combination of national and provincial models can help predict the current and future ecological spatial niche of the black stork. By combining these models, we can obtain a more comprehensive understanding of the underlying processes that shape the wintering habitat distribution of the black stork and identify the most critical areas for conservation efforts.

Overall, the aims of this study were: (i) to determine the extent to which climate and climate-driven LULC change will impact the wintering habitat of the black stork in Shaanxi province, (ii) to identify and quantify the effects of changes in climate and anthropogenic LULC on the suitability of the black stork in Shaanxi province, while considering local drivers, and (iii) to provide a scientific basis for the conservation of the black stork and other endangered birds.

## 2. Materials and Methods

### 2.1. Species Occurrence Data and Scales

#### 2.1.1. National Scale Database 

In this study, we considered the distribution of wintering sites of the black stork in continental China on a regional scale. To define the regional scale, we adopted the national extent proposed by El-Gabbas and Dormann, which is a geographic extent smaller than the entire range of the species but much larger than the landscape extent [26]. For the national scale, we collected wintering points of the black stork (only recordings between November and February when the black stork winters most frequently across its entire Chinese wintering range) from China Bird Report (http://www.birdreport.cn/, accessed on 16 December 2022) and China Animal Scientific Database (http://www.zoology.especies.cn, accessed on 18 December 2022). Since the localities were obtained from multiple unplanned surveys, these occurrence points are likely to exhibit spatial autocorrelation [27]. To reduce this autocorrelation, we filtered localities by randomly selecting one occurrence point within each 10 × 10 km^2^ grid, which is equal to the cell size of environmental variables at the national scale [1,12]. Finally, 174 records of black stork wintering sites were retained for models at the national scale (Figure 1 and Figure 2). 

#### 2.1.2. Provincial Scale Database

From 2016 to 2021, we conducted active searches for black storks in their wintering areas in Shaanxi province, mainly in the Yellow River, Hanjiang River, and Weihe River basins, and compiled GPS coordinates of wintering sites. We identified black storks in 55 locations, including river banks, pools, marshes, and rice fields, where at least one bird was recorded. In total, we recorded 366 wintering black storks in Shaanxi province. To reduce collinearity, we applied spatial filtering to the 55 occurrences. Our predictor variables for the provincial scale had a resolution of 300 × 300 m^2^, and in cases where multiple occurrence locations fell within a single grid cell, we kept only one record. After spatial filtering, 41 occurrence points remained for simulating provincial models (Figure 2).

### 2.2. Environmental Predictor Variables 

To calibrate the hierarchical SDMs, we utilized two datasets encompassing various environmental factors with different spatial resolutions at two specific scales. At the national scale, we used climate and LULC variables at a resolution of 10 km for the entire Chinese mainland. We employed LULC as a means to integrate significant predictors, including habitat type, vegetation structure, and food availability, into our modeling process [28,29]. The LULC types included evergreen broadleaf forest, deciduous broadleaf forest, evergreen needleleaf forest, deciduous needleleaf forest, mixed forest, shrub land, grasslands, wetlands, croplands, water bodies, snow and ice, built-up land, and barren or sparsely vegetated land. 

In order to capture plausible variations in future climate, we extracted projections of climate for 2030 and 2050 from the BCC-CSM2-MR Global Circulation Models (GCMs) used in the sixth assessment report of the Intergovernmental Panel on Climate Change (IPCC). We used the climate scenarios SSP1-2.6, SSP2-4.5, and SSP5-8.5 to create future projections, which are based on greenhouse gas emission and socioeconomic patterns according to the Coupled Model Intercomparison Project Phase 6 (CMIP6) [30,31]. Thus, we considered a more optimistic scenario (SSP126), a moderate scenario (SSP245), and a more pessimistic scenario (SSP585) for climate projections. Additionally, for each climate scenario, we used the corresponding simulated LULC data (see below). Climatic variables were obtained from WorldClim [32] with a spatial resolution of 2.5 min (http://www.worldclim.org, accessed on 22 September 2022), and Scenario datasets of land cover in Eurasia (2020–2050) were obtained from the Science Data Bank (https://www.scidb.cn/en, accessed on 22 September 2022), and resampled to a 10 km^2^ cell size for use in the national-scale models. All spatial procedures were implemented in ArcGIS Pro (https://www.esri.com/en-us/arcgis/products/arcgis-pro).

Provincial models were calibrated with local environmental variables at a precise resolution of 300 m to reflect the local conditions available for the wintering black stork. We generated different environmental variables, such as topographical variables, human influence variables, and water or food supply variables. To account for pairwise correlation between predictors, we subselected the final set of variables by considering Pearson’s correlation coefficient of |r| < 0.75. These procedures were carried out in the R [33] environment. Finally, we retained the five most uncorrelated and ecologically meaningful variables for the national models, namely, isothermality, precipitation of driest month, mean temperature of driest quarter, precipitation seasonality, and LULC. For the provincial models, we retained seven variables, including aspect, elevation, LULC, nightlight, slope, distance to the main road, and distance to water (Appendix A). 

### 2.3. LULC Simulation

In this study, we utilized the Patch-generating Land Use Simulation (PLUS) model [34] to simulate land use and land cover changes by integrating anthropogenic and natural factors. The PLUS model employs the Land Expansion Analysis Strategy (LEAS) and a Cellular Automata (CA) model based on multiple types of Random Seeds (CARS). This model is unique in its ability to identify drivers of land expansion and project landscape dynamics, and its simulation accuracy exceeds that of other models [35]. The PLUS model simulates LULC changes based on the interaction and competition between different land use types, as well as driving factors such as socioeconomic, climatic, and environmental factors that contribute to the dynamics of land type transition.

To simulate LULC changes in Shaanxi, we collected 13 natural and socioeconomic driving factors as auxiliary data, including population density, GDP, soil, DEM, annual mean temperature, annual precipitation, distance from highways, railways, primary and secondary roads, towns, county centers, and prefectural centers. Detailed information on these driving factors can be found in Appendix A. 

Firstly, we predicted the land use structure in 2020 based on the 2000 and 2010 LULC maps, and we used the actual 2020 LULC map to verify the accuracy of the model. Secondly, we simulated future LULC maps for 2030 and 2050, including six general (cropland, forest, grassland, wetland, impervious, and bare land) and 22 detailed land use types to be consistent with the climate and LULC data on the national scale. The simulation results obtained from PLUS were validated using two common measures, namely the Overall Accuracy (OA) [34] and the Kappa Coefficient [36], which are generally used to evaluate the level of agreement between two different sets of data images.

### 2.4. Species Distribution Modelling 

We calibrated SDMs for two scales: national and provincial. The national models were calibrated for the Chinese mainland using data from the national occurrence database and only macroclimatic and climate-related LULC variables at a resolution of 10 km. Provincial models were calibrated for the Shaanxi province using local environmental variables at a resolution of 300 × 300 m^2^ to predict the current ecological spatial niche pattern of the black stork. For each database, we generated an ensemble model by combining six statistical techniques: Boosted Regression Trees (BRT), Generalized Additive Models (GAM), Generalized Linear Models (GLM), Maximum Entropy (MXD), Radom Forest (RDF), and Support Vector Machines (SVM). All the models were generated using the ENMTML [37] R package. For each modeling algorithm, we randomly divided the corresponding data into two sets, with 80% of the data used for generating the models and 20% of the data used to estimate the predictive accuracy of each model. The variable importance was calculated with the imp_var function implemented in the ENMTML package. 

We evaluated each model using seven different metrics: Area Under the Curve (AUC) [38], Kappa [39], True Skill Statistic (TSS) [40], Jaccard [41], Sorensen [41], Fpb [42], and Boyce [43]. A higher value for each metric indicates greater accuracy of the model. To study the potential distribution of the black stork, we created a final model by ensembling all the algorithms using a weighted mean method [44]. This method averages the suitability values weighted by the performance of the algorithms, specifically the TSS value. The TSS values were used to determine the weight of each individual model, as shown in the following equation:(1)$$W_{j}=\frac{r_{j}}{\sum_{j=1}^{h}r_{j}}$$
where $W_{j}$ represents the weight of the model result j; $r_{j}$ is the TSS value of the model result j; and h is the number of the model results. 

To investigate the potential impacts of future climate and LULC changes on the suitable habitats of wintering black storks, we conducted a series of analyses using ArcGIS Pro (https://www.esri.com/en-us/arcgis/products/arcgis-pro). The first step involved classifying habitats into binary data based on the Maximum Training Sensitivity plus Specificity (MTSS) logistic threshold. Subsequently, a comparison was made between the distribution of wintering habitats in the current period and in the future projections for the 2030s and 2050s. Additionally, we identified climate refugia areas, which are currently climatically suitable and predicted to remain so under future climate conditions [45], by comparing the current climatically suitable map with the future maps.

## 3. Results

### 3.1. Predicted LULC Changes 

The results, with a Kappa coefficient of 0.83 and an OA of 0.88, suggest that the PLUS model has greater accuracy in simulating LULC changes and is capable of more precisely reflecting the actual variations in LULC in Shaanxi Province. As a consequence, we proceeded to choose the most suitable parameter sets to predict the LULC changes for both 2030 and 2050.

The LULC maps for the future, generated by the PLUS model, are presented in Figure 3. The simulation results reveal that if the transition rate remains the same as that of the past decade (i.e., 2010–2020), dramatic LULC changes are expected in the study area by 2030 and 2050, primarily along the Wudinghe River and in the Weihe and Hanjiang River basins (Figure 3a,b). By 2030, the area of cropland, forest, grassland, wetland, impervious, and bare land is projected to change by −1.53%, 0.71%, −1.87%, 0.71%, 34.69%, and 0.85%, respectively, compared to the year 2020; while by 2050, these are expected to change by 4.53%, 9.65%, 0.17%, 13.95%, 114.26%, and −2.46%, respectively (Table 1).

### 3.2. Major Environmental Factors That Affected Wintering Habitat Distribution

Table 2 presents the average measures for 10 runs to test the predictive performance of the national and provincial SDMs. The overall accuracy, in terms of averages, is 0.81 (national) and 0.82 (provincial). These results indicate that the distributions of wintering habitat of black storks in China or Shaanxi province are well described by the selected predictors, making them suitable for deriving future projections. 

Table 3 illustrates the importance of the predictors selected in this study. The mean temperature of the driest quarter (Bio9) and Precipitation of the driest month (Bio14) have the highest contribution to the national SDM, while Euclidian distance to water and LULC are the predictors with the strongest influence on the distribution of black stork’s wintering habitat in provincial level. The importance of temperature and precipitation for characterizing and delimiting the distribution of the black stork has previously been reported by Cano et al. (2014) [46]. As shown in Figure 4a, for the national model, when the Mean temperature of the driest quarter is >12.7 °C or <−6.3 °C, it is not suitable for the black stork. The highest suitability occurs around 0 °C. When the Precipitation of the driest month is <36.3 mm, the regions are not suitable for the black stork. There are two distinct peaks of precipitation suitability when the precipitation range falls between 42 and 78 mm. (Figure 4b). The provincial SDM results show that areas within a 2500 m radius of water bodies are highly suitable for the black stork to overwinter, while rivers, reservoirs, river shallows, and urban lands serve as more favorable LULC types for the wintering black stork (Figure 5).

### 3.3. National and Provincial Scale SDMs for the Current Scenario

Figure 6 shows the wintering habitat suitability of the black stork in China. Based on the ensemble model, the suitable regions (regions with a probability of presence greater than the MTSS threshold of 0.32) for the black stork were mainly in the middle and lower reaches of the Yangtze River and the Yellow River, accounting for approximately 22.7% of the area of China. 

The adjustment of the provincial models showed that by utilizing the MTSS thresholding method, 46.9% of the total area of Shaanxi province was projected to be potential wintering habitat. In this context, we define this portion of land as a locally suitable area. As expected, the distribution maps indicate distinct spatial patterns between the national and provincial models (Figure 7a and Table 4). Regarding Shaanxi province, national models, as hypothesized, predicted higher percentages of potential distribution than provincial models, reaching 61.8% of the area of Shaanxi province, which we refer to as a climatically suitable area (macroclimate spatial niche forecast).

The overall suitable areas, i.e., overlapping areas of the climatically and locally suitable regions, were mainly distributed in Hanjiang and Weihe River basins in south-central Shaanxi, as well as Hongjian Nur and Wuding River basins in northern Shaanxi, accounting for 29.8% of the area of Shaanxi province (Figure 8). 

### 3.4. Projections of Future Scenarios

The wintering habitat projections of the black stork in China varied among the three future scenarios (SSP 126, SSP 245, and SSP 585). The national models projected to the SSP 126 scenario showed a significant increase in the climatically suitable area in Shaanxi province, with increases of 7.7% and 6.9% in the 2030s and 2050s, respectively. Under the SSP 245 future scenario, the climatically suitable area projected for the 2030s and 2050s was 8.8% and 6.6% larger than the contemporary suitable area, respectively. Under the SSP 585 scenario, climatically suitable areas increased by 4.8% and 5.3% in the 2030s and 2050s, respectively. The predicted potential wintering habitat of the black stork from our national models would be expanding due to climate change (Table 4, Appendix A). When projected onto future climate scenarios, the wintering habitat of the black stork will expand northward, and the expansion appears more pronounced between 37°–38° N (Figure 9). In Shaanxi province, the current potential wintering habitat of the black stork is distributed from 31° N to 39° N and peaked around 34.1° N. Under different future scenarios, the latitude of the location of the peak in the overwintering habitat area expands northward in the range of 0.1 to 0.5 arc-degrees. Under the low emission scenario (SSP126), the projected habitat areas in the 2030s and 2050s peak at 34.4° N and 34.2° N, respectively, while under the high emission scenario (SSP585), the projected habitat areas in both the 2030s and 2050s peak at 34.6° N, suggesting that the accelerating climate change under higher emission scenarios will cause the wintering habitat area of the black stork to move farther north. 

Based on the results of the provincial model, the future wintering habitat of the black stork in both 2030 and 2050 would be affected by future LULC changes. Figure 7 shows the potentially suitable wintering habitat of the black stork under the current (2020) LULC condition and simulates the 2030s and 2050s LULC conditions predicted by the provincial model, i.e., locally suitable area. Under the influence of LULC change, the total potentially suitable wintering habitat area of the black stork in Shaanxi province decreased by 11.7% and 17.3% in the 2030s and 2050s, respectively (Table 4).

The climatically suitable areas and locally suitable areas were overlaid to generate three maps of overall habitat suitability (one for each scenario) for each period of interest. Thus, the local suitable areas outside the climatically suitable areas are not considered as the final suitable wintering habitat. Under the current condition, the final suitable habitat for wintering black stork was 77,938 km^2^, accounting for 37.90% of the total area in Shaanxi province (Table 4). Under the SSP126 scenario, the final suitable habitat area of the black stork decreased by 10.25% and 4.23% in the 2030s and 2050s, respectively (Table 4). For the SSP245 scenario, the final potentially wintering habitat area of the black stork did not significantly decrease but increased slightly in the 2050s. It will decrease by 2.80% in the 2030s and increase by 0.83% in the 2050s. In the SSP585 scenario, the final potentially wintering habitat area of the black stork decreased by 11.49% and 4.14% in the 2030s and 2050s, respectively (Table 4). In general, with climate change, the final potentially wintering habitat area for the black stork is the largest under the SSP245 scenario, 75,749 km^2^ and 77,587 km^2^, respectively, in the 2030s and 2050s, followed by SSP126 and SSP585 (Figure 8 and Table 4). 

In the provincial SDM, the future wintering habitat area will be decreasing under the influence of LULC changes in both the 2030s and 2050s. However, the stacked model of two different spatial scales showed that under the combined influence of climate and LULC changes, the final potential wintering habitat under all future scenarios showed a trend of first decreasing in 2030 and slowly increasing by 2050 (Table 5).

The climate refugia for wintering black storks were mainly distributed in the wetlands of the Weihe River basin, Yellow River basin, and Southern Loess Plateau, with an area of 57,965 km^2^ (Figure 10). There were major conservation gaps in the suitable wintering habitat of the black stork, as only 2.7% (2085 km^2^) and 0.4% (351 km^2^) of the current potential wintering habitat were within natural reserves and wetland parks, respectively. 

## 4. Discussion

Although future simulations involve numerous uncertainties, such studies are valuable in obtaining information about species distributions, range shifts, and habitat availability and aid in mitigation and adaptation planning. Simulations incorporating LULC changes, climate projections, and species distribution provide a comprehensive understanding of ecosystem processes and their responses to environmental changes. The human-induced changes in land systems and climate have significant consequences for natural systems [47]. These changes, such as habitat loss, species extinction, and other impacts, highlight the importance of comprehending global and regional trends for effective ecosystem management. 

The present study projected the wintering habitat distributions of a migratory wading bird in a very important biogeographic region of China, Shaanxi Province. Our modeling framework integrates climate and LULC data in a scale-dependent hierarchical manner and provides valuable insights into how climate and LULC influence the distribution of species. By using this approach, we are able to identify regions that have favorable climate conditions but unsuitable land cover, and make predictions about how species distribution could change under scenarios of changing climate and LULC types. We analyzed the possible impacts of climate and LULC on the distributions of wintering habitat at national and provincial scales, highlighting the importance of considering both factors in conservation planning. 

There were significant differences observed between the SDMs developed at national and provincial scales. For example, provincial SDMs may accurately represent local ecological conditions for the black stork but may not fully capture the species’ complete climatic niche [7,48]. Consequently, models fitted with data from a partial section of a species’ environmental niche can produce truncated response curves [49], leading to biased simulation results of distribution [48]. Conversely, national SDMs can encompass a greater portion of the species’ climatic niche compared to provincial SDMs. However, at smaller ranges, such as certain areas in Shaanxi province that are less than two hundred kilometers wide, there is a possibility of overestimating suitable wintering habitats for the black stork. Therefore, the predicted suitable wintering habitat area for the black stork was greater in the national SDMs compared to the provincial SDMs. These results align with our initial hypothesis regarding the differences in results based on different scales. Previous studies have highlighted that SDMs calibrated at smaller scales are limited by their incomplete representation of the species’ ecological niche, potentially leading to biased future projections [13,18,19,48]. Due to this consideration, projections under various climate change scenarios were exclusively generated at the national scale. 

During our study period, under the impact of climate change, areas of climatic suitability for wintering black storks expanded across both the entire China and Shaanxi province. The temperature of the driest quarter and precipitation of the driest month are the two climatic variables that have the greatest impact on the distribution of wintering black storks. December to February is the driest quarter in northwestern China [50], and January is the driest month in Shaanxi Province [51], during which the black storks overwinter in Shaanxi Province. Climate change characterized by climate warming will undoubtedly have a notable effect on wintering habitat distribution. Cano et al. (2014) [46] demonstrated that slight increases in temperature can be advantageous for the distribution of black storks. The phenomenon of global warming, particularly in regions located at intermediate and higher latitudes, may be the primary factor leading to the transformation of regions in northern Shaanxi from being unsuitable to becoming climatically suitable in future scenarios, especially under SSP245. Telleria et al. (2016) [22] showed that the occupation of warm areas can reduce the need for thermogenesis and thus reduce individual daily energy requirements, allowing more migratory birds to occur at higher altitudes and lower temperatures. Moreover, the warming climate will enhance the suitability of colder sectors, such as mountains and plateaus, leading to a patchy distribution of climate patterns in the future.

Previous studies conducted on changes in suitable habitats under climate change typically did not consider LULC [52,53] or only included it as a variable within the model [46,54]. In this study, we simulated the effects of LULC change on species distribution at appropriate scales, allowing us to extrapolate to future distribution patterns. Our approach directly took into account the spatial limitations imposed by anthropogenic LULC when evaluating the impacts of climate change [55,56]. By incorporating LULC changes, we imposed essential spatial constraints on the current and future climatic suitability of wintering habitats. This conservative approach prevented the overestimation of species’ future distribution, thus ensuring that the conservation priority of species was not underestimated [57,58,59]. Results of provincial SDMs suggest that water resources (wetlands) and LULC are the two major influencing factors, with waters within 2500 m and wetland types such as rivers, reservoirs, and riverbanks being suitable areas for overwintering. Birds rely on specific habitats, such as forests or wetlands, for breeding and wintering. As a result of human disturbance, these habitats may shift or disappear, causing birds to either adapt or migrate to new areas [22]. Wintering black storks will face negative outcomes associated with LULC changes in the form of habitat range reduction by 2030 and 2050. These findings align with previous studies that suggest distribution ranges of certain species may suffer a local reduction or even complete loss due to changes in land cover [4,60,61,62]. 

The final SDMs derived from hierarchical combinations of national and provincial models showed that the combination of climate change and LULC change had slighter effects on changing the wintering habitat distribution areas than those with LULC change only [63]. Under the influence of LULC changes only, the suitable wintering habitat area will keep decreasing until 2050, while under the combined influence of climate and LULC changes, it will decrease first in 2030 and slowly increase in 2050, with the SSP245 scenario being the most obvious (Appendix A). These findings are highly consistent with the suggestion made by [64] that climate change modifies the risk of global biodiversity and habitat losses due to anthropogenic LULC change. The possible reason climate change can mitigate the risk of habitat losses due to anthropogenic LULC change is that it can cause changes in temperature and precipitation that enable species to expand their geographic range, potentially offsetting some of the negative impacts of habitat due to LULC change. For example, in some areas, warming temperatures have allowed species to expand their ranges into regions where they were not previously found. However, it should be noted that climate change also poses a significant threat to biodiversity and habitat distribution in its own right in most cases. The potential for climate change to mitigate the effects of LULC change should not be seen as a license to continue with unsustainable land use practices. Instead, it highlights the need for an integrated approach to conservation that addresses both the direct and indirect impacts of human activities on wild habitats. 

In China, nature reserves play a crucial role as the most significant and effective refuge for wild species [65]. However, more than 95% of the climatic refuges for wintering black storks fall outside the natural reserves or wetland park network. Although the main protection target of these reserves is not the wintering black stork, its migration route coincides with the East Asia-Australia flyway within the territory of Shaanxi. Shrinking the protection gaps will provide better conditions for the wintering black stork as well as other waders such as common cranes (*Grus grus*), Eurasian spoonbills (*Platalea leucorodia*), and Oriental storks (*Ciconia boyciana*) to overwinter. 

The suitable wintering habitat areas identified by our model, particularly those expected to remain suitable under climate and LULC changes (such as the Weihe River basin and its tributaries, as well as the Yellow River basin), are ideal choices for concentrating conservation efforts for habitat management and monitoring of wintering populations of the black stork. Further investigations into the distribution and characteristics of breeding and wintering habitats for the black stork in other provinces of China are necessary to investigate the effectiveness of protected areas and to ensure that the seasonal needs of this species are met throughout China. 

## 5. Conclusions

This study is the first hierarchical assessment to examine the potential effects of future climate and LULC changes on the wintering habitat distribution of the black stork in Shaanxi province, China. The complexity of the habitat selection process for migratory birds makes it challenging to provide an accurate description. The findings of this study can be seen as a representation of the potential presence or absence of migratory bird species in relation to environmental factors. Our study has demonstrated that the combined effects of future climate and LULC changes can have significant impacts on the distribution of wintering habitats. It is important to highlight that climate change seems to be lessening the negative impact on black stork wintering habitats caused by human-induced changes in land use and land cover. This suggests that wildlife is gradually adapting to the gradual changes in climate. However, it is more difficult for them to adapt to the short-term damage caused by human activities. Additionally, the wintering habitat of the black stork is not confined solely to nature reserves in Shaanxi Province. It is spread across a wider area that is subject to various human activities. This reminds us to allocate more resources toward safeguarding the black stork’s habitat beyond the boundaries of protected areas.

Given that black storks migrate long distances across the world and exhibit different habitat preferences during the wintering and breeding seasons, understanding the complex interactions between climate and LULC is crucial for effective conservation strategies.

## Figures and Tables

**Figure 1 animals-13-02726-f001:**
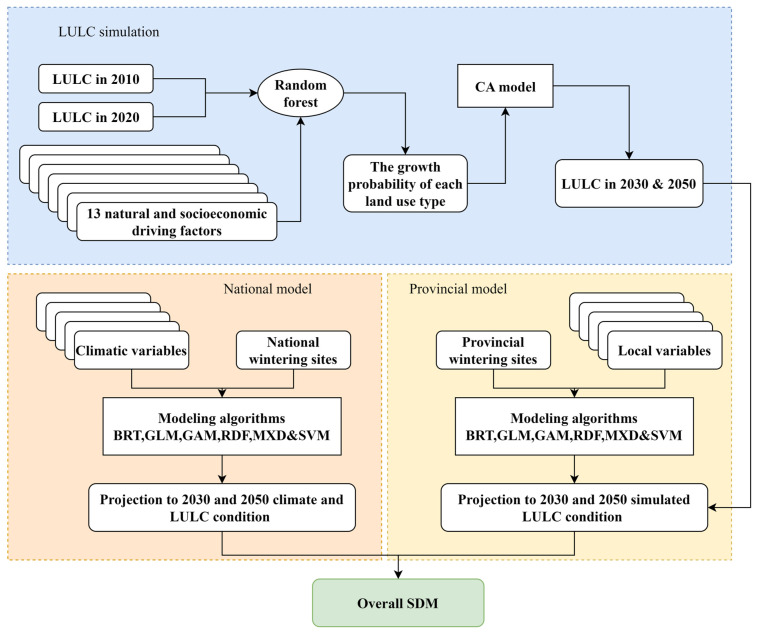
Framework of the patch-generating land use simulation model and the species distribution modeling workflow.

**Figure 2 animals-13-02726-f002:**
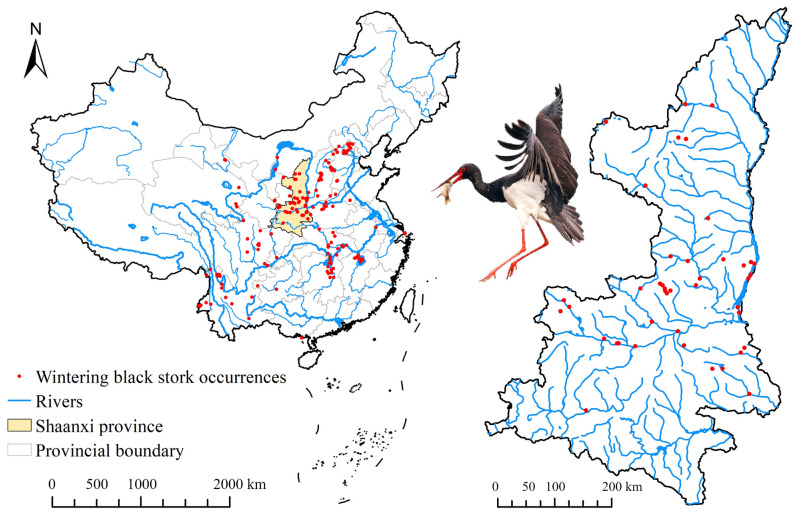
Occurrence points of wintering black stork in the China and Shaanxi provinces.

**Figure 3 animals-13-02726-f003:**
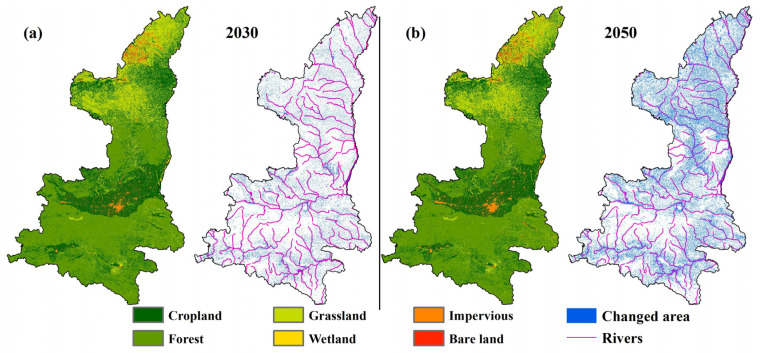
The simulated LULC changes show (**a**) the future LULC pattern for 2030 and LULC change between 2020 and 2030, and (**b**) the future LULC pattern for 2050 and LULC change between 2020 and 2050.

**Figure 4 animals-13-02726-f004:**
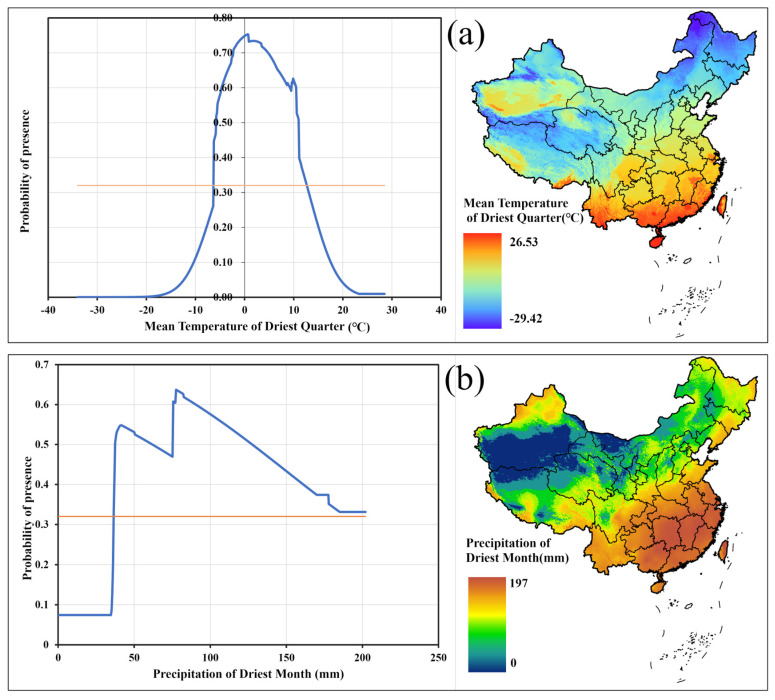
Important influencing factors for the national model: (**a**) Mean temperature of driest quarter, and (**b**) Precipitation of driest month (The blue curve represents the probability of presence and the orange line represents the MTSS threshold).

**Figure 5 animals-13-02726-f005:**
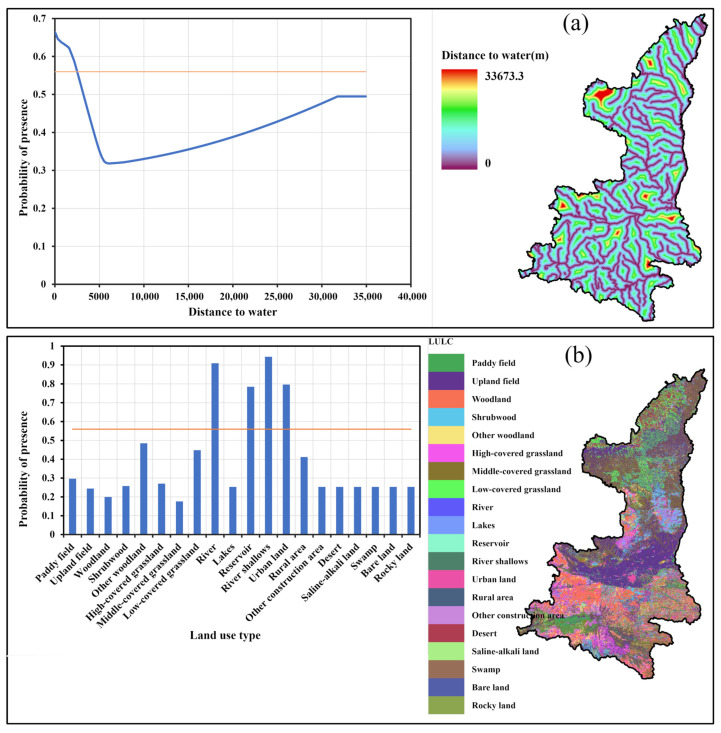
Important influencing factors for the Provincial model: (**a**) distance to water, and (**b**) LULC (The blue curve and histogram bars represent the probability of presence and the orange line represents the MTSS threshold).

**Figure 6 animals-13-02726-f006:**
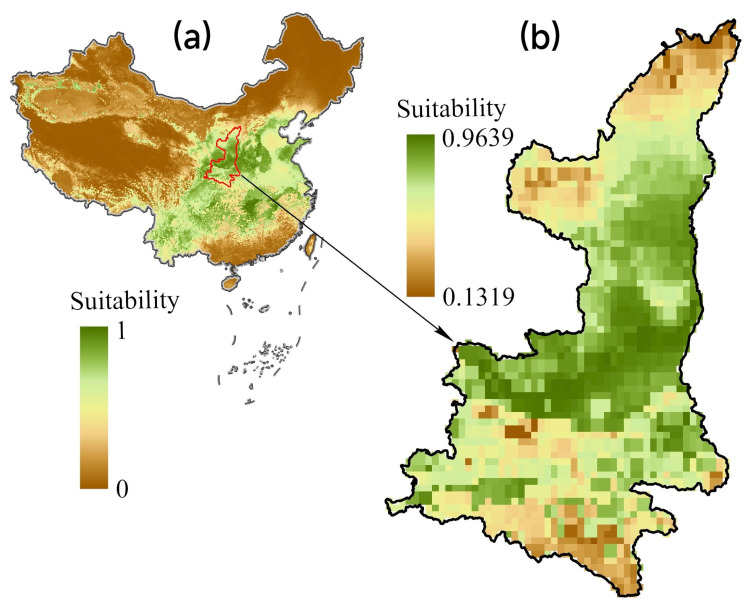
Current habitat suitability derived from national SDM for wintering black stork, (**a**) distribution pattern in China, and (**b**) distribution pattern in Shaanxi province.

**Figure 7 animals-13-02726-f007:**
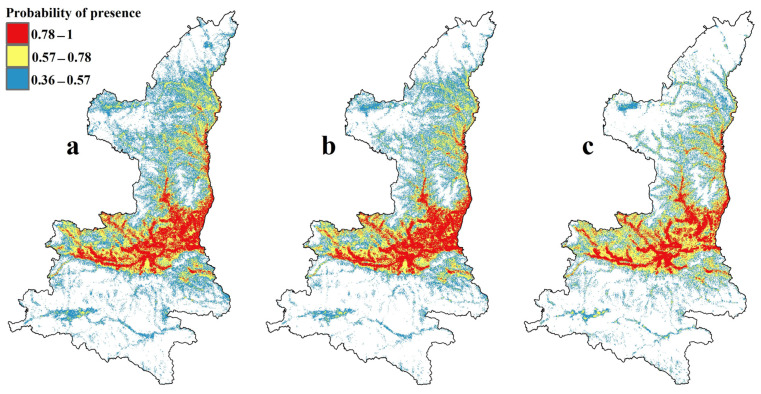
Current and future habitat suitability derived from provincial SDM for wintering black stork, (**a**) current distribution based on baseline LULC, (**b**) future distribution for 2030 based on simulated 2030 LULC, and (**c**) future distribution for 2050 based on simulated 2050 LULC.

**Figure 8 animals-13-02726-f008:**
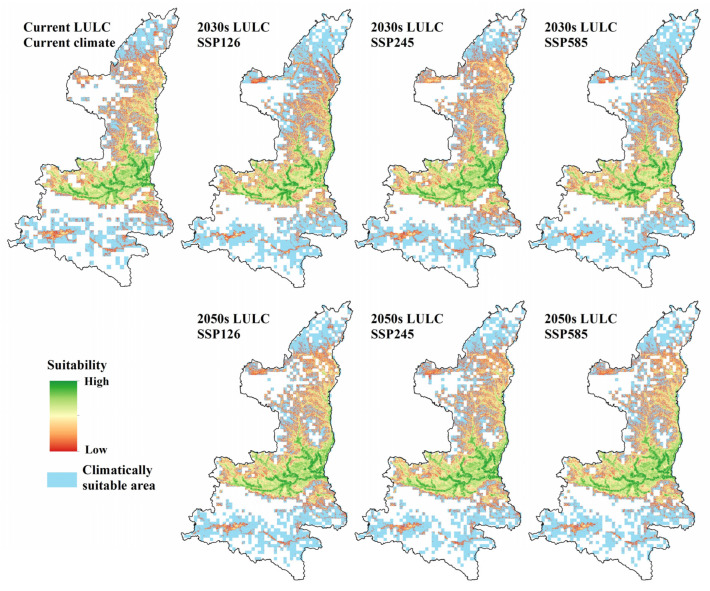
Final suitability maps for each climate and LULC change scenarios (the lower limit of the color band is the MTSS value; no unsuitable habitat is included on the maps; warmer colors indicate low suitability, and cooler colors indicate high suitability).

**Figure 9 animals-13-02726-f009:**
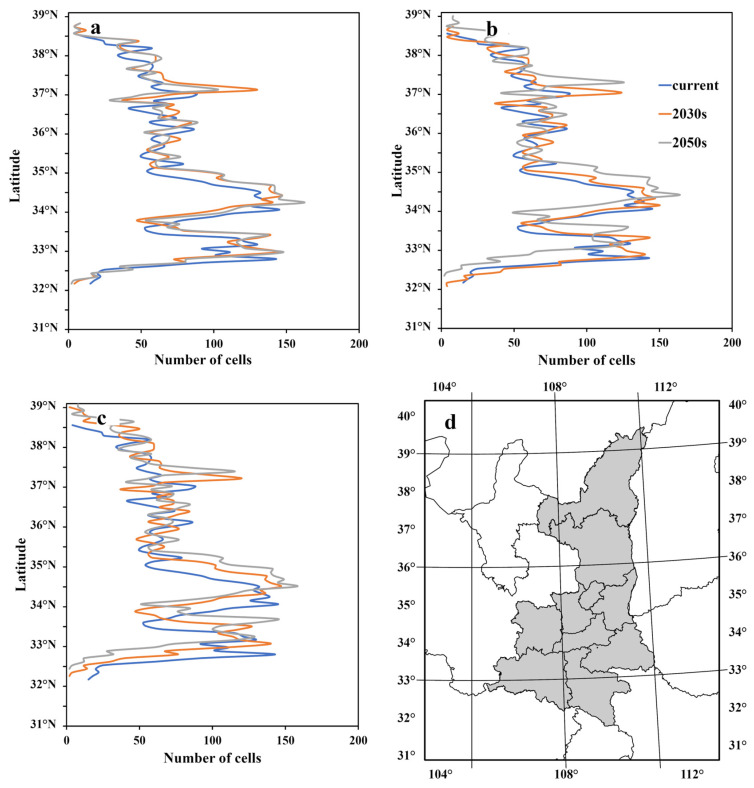
Latitudinal pattern of wintering black stork area (bandwidth = 0.1 degree), (**a**) SSP126, (**b**) SSP245, (**c**) SSP585, and (**d**) Location of Shaanxi province.

**Figure 10 animals-13-02726-f010:**
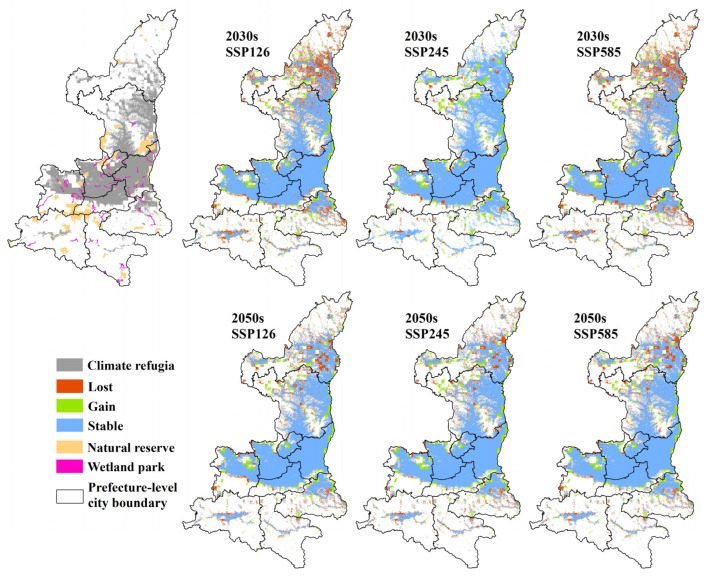
Climate refugia for wintering black stork and dynamics of habitat area under future scenarios (the top left map contains the distribution of climate refuges and protected areas, and the other scenarios show the habitat dynamics for the corresponding period).

**Table 1 animals-13-02726-t001:** Summary statistics of LULC patterns and their changes between 2020 and 2050.

	Year 2020	Year 2030	Change in 2020–2030 (%)	Year 2050	Change in 2020–2050 (%)
Cropland	59,858.55	58,964.04	−1.53	62,569.8	4.53
Forest	83,945.79	84,545.82	0.71	92,049.57	9.65
Grassland	42,187.5	41,398.74	−1.87	42,259.5	0.17
Wetland	1018.08	1025.28	0.71	1160.19	13.95
Impervious	3074.76	4141.26	34.69	6588.27	114.26
Bare land	1436.04	1448.19	0.85	1400.67	−2.46

**Table 2 animals-13-02726-t002:** Summary of measures for evaluating metrics calculated using the training dataset.

Scale	Metrics					
	AUC	Kappa	TSS	Jaccard	Sorensen	Boyce
National	0.91	0.69	0.69	0.75	0.85	0.96
Provincial	0.93	0.78	0.78	0.82	0.90	0.73

**Table 3 animals-13-02726-t003:** Importance of the environmental variables used to model the habitat suitability of wintering black stork.

Scale	Variable	Algorithms						μ (Mean)
		MXD	GLM	GAM	RDF	SVM	BRT	
National	LULC	0.13	0.30	0.14	0.15	0.04	0.11	0.14
	Bio15	0.17	0.16	0.18	0.18	0.16	0.12	0.16
	Bio3	0.03	0.07	0.09	0.18	0.04	0.14	0.09
	Bio14	0.27	0.15	0.22	0.20	0.31	0.14	0.21
	Bio9	0.40	0.33	0.37	0.29	0.46	0.49	0.39
Provincial	Aspect	0.01	0.04	0.01	0.07	0.02	0.03	0.03
	Elevation	0.07	0.07	0.13	0.14	0.09	0.06	0.09
	LULC	0.14	0.26	0.18	0.15	0.14	0.14	0.17
	Nightlight	0.29	0.04	0.04	0.13	0.16	0.08	0.13
	Dist_road	0.11	0.15	0.14	0.13	0.12	0.05	0.12
	Slope	0.19	0.14	0.05	0.18	0.20	0.14	0.15
	Dist_water	0.19	0.30	0.47	0.21	0.27	0.50	0.32

**Table 4 animals-13-02726-t004:** Projected changes in habitat suitability (% of the currently suitable area) for wintering black stork under climate and LULC change scenarios generated by two different spatial scales (km^2^).

	National Model			Provincial Model		
	Current	2030s	2050s	Current	2030s	2050s
LULC change only				96,333	85,006 (−11.7%)	79,698 (−17.3%)
Baseline condition	127,056			77,939		
SSP126		136,800 (7.6%)	135,800 (6.9%)		69,944 (−10.25%)	74,645 (−4.23%)
SSP245		138,300 (8.8%)	135,400 (6.6%)		75,749 (−2.80%)	78,588 (0.83)
SSP585		133,200 (4.8%)	133,800 (5.3%)		68,983 (−11.49%)	74,709 (−4.14%)

**Table 5 animals-13-02726-t005:** Predicted changes in final wintering habitat area for the black stork under climate and LULC change scenarios (km^2^).

	2030s			2050s		
	Stable	Gain	Lost	Stable	Gain	Lost
SSP126	61,189	8754	16,749	66,432	8213	11,506
SSP245	73,322	10,265	4615	67,030	8718	10,908
SSP585	60,619	8363	17,319	66,408	8300	11,529

## Data Availability

Not applicable.

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
