# Peer review of "Modeling the Wintering Habitat Distribution of the Black Stork in Shaanxi, China: A Hierarchical Integration of Climate and Land Use/Land Cover Data"

_animals, 2023, doi:10.3390/ani13172726_

Round 1
Reviewer 1 Report
Dear Authors
Well done. The paper is much interestin and well planned. I would like to share with you, as well as suggest some changes in the text. I will describe them line by line:
5, 6-10 - close to surnames you put a, b,c... whereas close to affiliation - numbers. You should standardize it.
35-37 - each keyword I would write in small letter
From 39 and whole text - unfortunately, you did not cite papers in the proper way - you should read requirements of MDPI
39-48 - Does author Li et al is the only one team that write about it?
67 - Ciconia nigra (Linnaeus, 1758)
132 - [2018]
137 - the hyperlink should be black and written with the same font and size
154 - e g. 300x300m2, should be written 300x300 m2
157 - Figure one - change 2,000 km in the cale into 2000 km
179-180 -one each link is written in a different why? With hyperlink and without it?
190-193, 210-213 - all variables I would recommend to write with a small letters
227 - 300 m2
316, 320 - Fig. 8, Table 4
387 - please re-arrange this table - don't loose ')' alone in the fourth column
436, 458, 471-472- double unnecessary dot
Author Response
Dear Reviewer,
We feel great thanks for your professional review work on our article entitled “Modelling the wintering habitat distribution of the Black Stork in Shaanxi, China: a hierarchical integration of climate and land use/land cover data”. As you are concerned, there are several problems that need to be addressed. According to your nice suggestions, we have made extensive corrections to our previous draft, changes are marked in red fonts in the revised version of the manuscript, and our point-to-point responses to your comments are listed below.
- 5, 6-10 - close to surnames you put a, b,c... whereas close to affiliation - numbers. You should standardize it.
Response: We are very sorry for this mistake and we have corrected it.
- each keyword I would write in small letter
Response: We have corrected it.
- From 39 and whole text - unfortunately, you did not cite papers in the proper way - you should read requirements of MDPI
Response: As requested we have modified the reference formatting.
- 39-48 - Does author Li et al is the only one team that write about it?
Response: We sincerely appreciate the valuable comments. We have checked the literature carefully and added more references on global change in the Introduction part of the revised manuscript.
- 67 - Ciconia nigra (Linnaeus, 1758)
Response: Thanks for the correction, we've fixed it.
(6) 137 - the hyperlink should be black and written with the same font and size
Response: Thanks for the correction, we've fixed it.
(7) 154 - e g. 300x300m2, should be written 300x300 m2
Response: We are very sorry for this mistake and we have corrected it.
(8) 157 - Figure one - change 2,000 km in the cale into 2000 km
Response: We agree with your comments, but another reviewer suggested putting thousand separators on all numbers, and for the sake of consistency, this version was not changed here.
(9) 179-180 -one each link is written in a different why? With hyperlink and without it?
Response: We are very sorry for this mistake and we have corrected it.
(10) 190-193, 210-213 - all variables I would recommend to write with a small letters
Response: Thank you for your comments, we have corrected it.
(11) 387 - please re-arrange this table - don't loose ')' alone in the fourth column
Response: Thank you for your comments, we have rearranged this table.
(12) 436, 458, 471-472- double unnecessary dot
Response: We are very sorry for this mistake and we have corrected it.

Reviewer 2 Report
The paper is an interesting exercise of modeling the wintering distribution of a bird species and the potential effects of climate change on it. It is particularly interesting because, contrary to other studies, models the change of land cover and land uses in the future and incorporates these predicted changes to the modeling of wintering distribution of the black stork. However, I believe that this part needs more detailed and clear explanation in the methods section, as I explain in the comments below. I believe that some parts of discussion should be improved and particularly the conclusion section would need a thorough revision.
References do not fit to the journal format and have some mistakes.
L159: It would be better to call this section “environmental predictor variables”
L178: Add after LULC data “(see below)” to make clear to the reader that these simulations are explained later in the text
L179-180: specify which data sets were used since the web cited includes a huge number of data sets
L188: Add reference for R
L194-197: it is strange to see hypotheses described in the methods section
L220: Add references for OA and Kappa
L279-282. Figure 3. It would be helpful if you could add the current LULC map to the figure and if pixels in the changed area map could be assigned different colors based on the specific predicted changes, such as forest to cropland, grassland to cropland, and so on. Alternatively, you could differentiate a selection of the most significant changes with distinct colors, while using a common color for the remaining changes.
L264-265: substitute projected or expected to increase by “to change”
L275: Table 2. Are these metrics calculated using the training data set (80% of the data) or are they computed with the remaining 20% data for model evaluation?
L270-271,275: Table 2. Overall accuracy has not been defined before, only mentioned in L220 without citing references. In this table Overall accuracy seems to be (and coincides with) the average of the metrics in that table. However, these metrics evaluate different characteristics of the models, thus it is nonsense calculate its average.
L297. Table 3. I have not found in the methods text or in the table text how the importance of the variables was calculated. I’m not sure if the means presented in the last column make sense.
L291. It is not logical to give an exact value; it would be better to say around 0 degrees. It would be important to know which is the driest quarter. Does it overlap with the wintering season of the stork?
L292-293: It is not logical to provide an exact value with two decimal positions for precipitation of the driest month. Furthermore, the graphic for precipitation is strange with two maximums. It would seem more logical to indicate the range of precipitations delimited by these two maximum values.
L293-294. Please mention the variable you are talking about (Distance to water)
L307: “probability of presence > 0.32” based on MTSS threshold I guess. I believe it would be nice to remind the reader this.
L314: “the percentage of potential wintering habitat area predicted was 46.9% of the area of Shaanxi”, based on MTSS threshold? Please clarify
L310-12: Authors write “national models, as hypothesized, predicted higher percentages of potential distribution than provincial models, reaching 61.8% of the area of Shaanxi province” but the results of provincial models are presented in the next paragraph, thus this sentence is somewhat out of place.
L325: Figure 7. Which were the criteria to determine the three intervals of probability of presence? Would it be possible to overlap a binary distribution based on MTSS thresholds onto these maps?
L387: Table 4. Figures in this table, are for the province only? Please clarify this. Numbers would be easier to understand if a thousand separator is added
L329: Figure 8. Would it be possible to overlay a binary distribution based on MTSS thresholds on these maps?
L364, 372 etc: Add thousand separators
L389. Table 5. Gains and losses compared to what? Gain and losses in 2050 are calculated compared to 2030 o compared to current distribution.
L393. Figure 10. I believe this figure requires more explanation. Does the top left map show the current situation? These maps would also benefit from overlaying the predicted binary distribution based on MTSS thresholds. Lost areas (red pixels) are lost compared to which distribution?
Furthermore, I suggest providing clearer explanations in the methods section for the comparisons of the extent of current and future distributions, gain and loss of areas, climatic refugia, etc. For instance, the concept of climatic refugia is first mentioned in the Results section, specifically in line 380.
L402: Venter et al. 2016 is lacking in references.
L405-436: The first part of the discussion largely repeats concepts already presented in the introduction, with a loose connection to the results of the study
L438-439: A discussion of the possible causal connections of the temperature of the driest quarter and the precipitation of the driest month with the distribution of the stork is needed. I guess that the driest quarter occurs in winter, but this should be explicitly stated somewhere to facilitate readers not familiarized with the region to understand results and discussion. There are some general comments on the effect of temperature and precipitation (L482-483) but they are not directly connected to the results of this study.
L514-516, L522-527: Conclusions should be improved as they include some sentences that hardly may be considered as such, for instance “Nevertheless, it is crucial to recognize that a model is a simplified representation of reality and only considers specific aspects of ecological systems …”. In the current version, this section contains overly general statements that lack a connection with the results of this study and could have been written without conducting it.
as far as I can see as a non native english speaker I believe english is OK
Author Response
Dear Reviewer,
We feel great thanks for your professional review work on our article entitled “Modelling the wintering habitat distribution of the Black Stork in Shaanxi, China: a hierarchical integration of climate and land use/land cover data”. As you are concerned, there are several problems that need to be addressed. According to your nice suggestions, we have made extensive corrections to our previous draft, changes are marked in red fonts in the revised version of the manuscript, and our point-to-point responses to your comments are listed below.
- References do not fit to the journal format and have some mistakes.
Response: We have revised the manuscript according to the reviewer's suggestions, specifically addressing the formatting of references and correcting some errors in the text.
- L159: It would be better to call this section “environmental predictor variables”
Response: Thank you for your comments, we have corrected it.
- L178: Add after LULC data “(see below)” to make clear to the reader that these simulations are explained later in the text
Response: Thanks for your careful checks. We are sorry for our carelessness, and we have corrected it.
- L179-180: Specify which data sets were used since the web cited includes a huge number of data sets
Response: We are very sorry for this mistake and we have specified datasets in the current version.
- L188: Add reference for R
Response: Thanks for your careful checks. We have added a reference for R.
- L194-197: it is strange to see hypotheses described in the methods section
Response: We thank you for the critical comments and helpful suggestions. As you suggested, it was indeed inappropriate and we have removed these redundancies.
- L220: Add references for OA and Kappa
Response: Thanks for your careful checks. We have added the appropriate references.
- L279-282. Figure 3. It would be helpful if you could add the current LULC map to the figure and if pixels in the changed area map could be assigned different colors based on the specific predicted changes, such as forest to cropland, grassland to cropland, and so on. Alternatively, you could differentiate a selection of the most significant changes with distinct colors, while using a common color for the remaining changes.
Response: Thanks for your careful checks. We have revised Figure 3 based on the suggestion.
- L264-265: substitute projected or expected to increase by “to change”
Response: Thanks for the correction, we've fixed it.
- L275: Table 2. Are these metrics calculated using the training data set (80% of the data) or are they computed with the remaining 20% data for model evaluation?
Response: Thanks for your careful checks. Metrics are calculated using the training data set. We make the appropriate notes in the table header.
- L270-271,275: Table 2. Overall accuracy has not been defined before, only mentioned in L220 without citing references. In this table Overall accuracy seems to be (and coincides with) the average of the metrics in that table. However, these metrics evaluate different characteristics of the models, thus it is nonsense calculate its average.
Response: We think this is an excellent suggestion. We sincerely appreciate your valuable comment. OA on L220 is a metric for PLUS model accuracy tests, not the same as Overall Accuracy here. It's a metric we came up with ourselves and simply averaged. As you suggested, it is obviously quite illogical, so the metric was removed from the article.
- Table 3. I have not found in the methods text or in the table text how the importance of the variables was calculated. I’m not sure if the means presented in the last column make sense.
Response: Thank you for pointing out this problem. As you suggested, we added the calculation of variable importance in the Materials and Methods section. According to the instructions of the ENMTML package, the variable importance of each independent algorithm is in the middle of 0-1, and the higher it is, the more important it is, so we referred to the previous similar literature, and averaged the importance values as the final indicator.
- It is not logical to give an exact value; it would be better to say around 0 degrees. It would be important to know which is the driest quarter. Does it overlap with the wintering season of the stork?
Response: Thanks for your careful checks. We are sorry for our carelessness. Based on your comments, we have made the correction to this point. December to February is the driest quarter in northwestern China, and January is the driest month in Shaanxi Province, during which the black storks overwinter in Shaanxi Province. In the current version, we have added the appropriate notes and referenced the corresponding literature.
- L292-293: It is not logical to provide an exact value with two decimal positions for precipitation of the driest month. Furthermore, the graphic for precipitation is strange with two maximums. It would seem more logical to indicate the range of precipitations delimited by these two maximum values.
Response: Thanks very much for your comment, which is highly appreciated. Based on your comments, we have modified our interpretation of the graphic.
- L293-294. Please mention the variable you are talking about (Distance to water)
Response: We are very sorry for this mistake and we have mentioned it in the text.
- L307: “probability of presence > 0.32” based on MTSS threshold I guess. I believe it would be nice to remind the reader this.
Response: We are sorry for our carelessness. Based on your comments, we have made the correction to this point.
- L314: “the percentage of potential wintering habitat area predicted was 46.9% of the area of Shaanxi”, based on MTSS threshold? Please clarify
Response: Yes, it is based on the MTSS threshold. This threshold is indicated on the corresponding tables and figures.
- L310-12: Authors write “national models, as hypothesized, predicted higher percentages of potential distribution than provincial models, reaching 61.8% of the area of Shaanxi province” but the results of provincial models are presented in the next paragraph, thus this sentence is somewhat out of place.
Response: Thank you for reading the article carefully, we followed the comments to enhance the article.
- L387: Table 4. Figures in this table, are for the province only? Please clarify this. Numbers would be easier to understand if a thousand separator is added
Response: The main focus of the article is the black stork habitat in Shaanxi Province, and the national scale is used in order to extract the climatic requirements of the black stork in Shaanxi Province and to find out the climatically suitable areas. Therefore, we didn't think it was necessary to compare the changes in the national scale habitat for different scenarios. This is written in the discussion part of the article.
- L329: Figure 8. Would it be possible to overlay a binary distribution based on MTSS thresholds on these maps?
Response: We thank you for the critical comments and helpful suggestions. The areas shown with color on the map are all areas greater than the MTSS threshold, and there are no unsuitable wintering habitats on the map. This is actually supposed to be a binary distribution map, which has the advantage of showing both suitable and unsuitable areas, as well as areas of high suitability with color bands. We labeled the map notes with this for Figure 8.
- L364, 372 etc: Add thousand separators
Response: Thanks for the correction, we've fixed it.
- Table 5. Gains and losses compared to what? Gain and losses in 2050 are calculated compared to 2030 o compared to current distribution.
Response: Thank you for pointing out this problem. Gain and losses are calculated compared to the current distribution. As you suggested, we have added a corresponding clarification in the methods section.
- Figure 10. I believe this figure requires more explanation. Does the top left map show the current situation? These maps would also benefit from overlaying the predicted binary distribution based on MTSS thresholds. Lost areas (red pixels) are lost compared to which distribution?
Response: Thank you for pointing out this problem. As you requested, we have included an explanation at the bottom of Figure 10. Like the previous figure, this figure has a binary distribution in it. the lost area is calculated by comparing it to the current one.
- Furthermore, I suggest providing clearer explanations in the methods section for the comparisons of the extent of current and future distributions, gain and loss of areas, climatic refugia, etc. For instance, the concept of climatic refugia is first mentioned in the Results section, specifically in line 380.
Response: Thank you for your valuable advice. As you suggested, we have added a corresponding clarification in the Methods section.
- L405-436: The first part of the discussion largely repeats concepts already presented in the introduction, with a loose connection to the results of the study. L438-439: A discussion of the possible causal connections of the temperature of the driest quarter and the precipitation of the driest month with the distribution of the stork is needed. I guess that the driest quarter occurs in winter, but this should be explicitly stated somewhere to facilitate readers not familiarized with the region to understand results and discussion. There are some general comments on the effect of temperature and precipitation (L482-483) but they are not directly connected to the results of this study.
Response: Thank you for your comments and suggestions concerning our discussion section. The comments and suggestions are all valuable and very helpful for revising and improving our paper, as well as the important guiding significance to our research. We have studied comments carefully and have made correction which we hope meet with approval.
- L514-516, L522-527: Conclusions should be improved as they include some sentences that hardly may be considered as such, for instance “Nevertheless, it is crucial to recognize that a model is a simplified representation of reality and only considers specific aspects of ecological systems …”. In the current version, this section contains overly general statements that lack a connection with the results of this study and could have been written without conducting it.
Response: We thank you for the critical comments and helpful suggestions. We have taken all these comments and suggestions into account, and have made major corrections in Conclusion section.
